# Reply to Gomez-Rojas, M.A.; Tung-Chen, Y. Comment on “Vicente-Rabaneda et al. A Proposal for a New Lung Ultrasound Score in Rheumatoid Arthritis: The Reliability of Lung Ultrasound for Rheumatoid Arthritis-Associated Interstitial Lung Disease Diagnosis. *J. Clin. Med.* 2025, *14*, 3701”

**DOI:** 10.3390/jcm14228119

**Published:** 2025-11-17

**Authors:** Esther Francisca Vicente-Rabaneda, Ingrid Möller, Abdon Mata, Nuria Montes, Gabriel-Santiago Rodríguez-Vargas, Luis Coronel, David Bong, Santos Castañeda, Pedro Santos-Moreno

**Affiliations:** 1Rheumatology Department, Hospital Universitario de La Princesa, IIS-Princesa, C/Diego de León 62, 28006 Madrid, Spain; nuria.montes.casado@gmail.com (N.M.); scastas@gmail.com (S.C.); 2Facultad de Medicina, Universidad Autónoma de Madrid, C/del Arzobispo Morcillo 4, 28029 Madrid, Spain; 3Instituto Poal de Reumatología, Carrer de Castanyer, 15, Sarrià-Sant Gervasi, 08022 Barcelona, Spain; ingrid.moller@ipoal.com (I.M.); dabong47@gmail.com (D.B.); 4Fundación Neumológica Colombiana, 13b Street #161-85, Bogotá 110221, Colombia; abdonmata@gmail.com; 5Research Department, Biomab IPS, 48th Street #13-86, Bogotá 110221, Colombia; santiago.rv18@gmail.com; 6Rheumatology Department, Barcelona Campus, Hospital Universitari Vall d’Hebron, 08035 Barcelona, Spain; luiscoroneltarancon@gmail.com; 7Bellvitge Campus, Universitat de Barcelona, Carrer de la Feixa Llarga s/n, L’Hospitalet de Llobregat, 08036 Barcelona, Spain; 8Rheumatology and Research Departments, Biomab IPS, 48th Street #13-86, Bogotá 110221, Colombia

We sincerely appreciate the interest and insightful comment [1] regarding our recent publication in the *Journal of Clinical Medicine* [2]. Like our colleagues, we are excited about the growing role of bedside lung ultrasound (LUS) in diagnosing interstitial lung disease (ILD) in patients with rheumatoid arthritis (RA) [1,2]. Their remarks provide valuable opportunities to improve the interpretation and clinical use of our findings.

Gomez-Rojas et al. expressed concerns that including a systematic literature review might affect the external validity of our study [1]. We respectfully disagree, as the main purpose of the literature review was to gather and evaluate existing evidence on the reliability of LUS in screening or diagnosing RA-ILD. This process allowed us to compare our findings with current knowledge and identify research gaps. Notably, our review found that intra- and inter-explorer reliability of LUS in RA-ILD has seldom been studied, with only one study providing data on inter-explorer reliability [3].

The comment also highlights potential confounders, which we addressed in the limitations section of our pilot study. Given its small sample size, caution is warranted when generalizing our conclusions. Nonetheless, a primary strength of our work is its focus on intra- and inter-examiner reliability, rather than observer reliability, providing added value to the field. Organizing such a study—where patients are examined twice by three trained examiners on the same day using identical ultrasound equipment—requires meticulous planning, limiting the feasible sample size to ensure proper blinding.

Regarding patient symptoms, our recruitment targeted individuals with suspected RA-ILD based on “Velcro” crackles, unexplained respiratory symptoms, abnormal lung function tests (LFTs), or imaging results. In such cases, screening for ILD is justified and cost-effective, performed as part of routine monitoring through clinical examination, LFT, and high-resolution computerized tomography (HRCT). LUS was incorporated as a point-of-care assessment for patients with chest symptoms suggestive of RA-ILD, in accordance with recent international guidelines [4]. Importantly, as reflected in Supplementary Table S1, some of the enrolled patients were asymptomatic, with suspicion arising from abnormal LFTs or imaging, which were performed due to the presence of risk factors of ILD. We would like to emphasize that our pilot study was designed to examine the precision—not accuracy—of LUS, which limited analysis of disease severity. However, given the favorable findings of our pilot study, we believe it would be of great interest to conduct further research with larger samples to validate our findings and compare LUS reliability across varying disease stages and in other rheumatic diseases. Nevertheless, the presence of RA-ILD was confirmed or excluded in all patients by HRCT. Additionally, the diagnostic performance of the proposed LUS scores was evaluated, showing sensitivity, specificity, cut-offs and area under the receiver operating characteristic curve (AUROC) in Table 5.

We concur that risk factors for RA-ILD must be considered, as recently mentioned in an expert proposal based on Delphi methodology performed among Spanish pulmonologists and rheumatologists [5]. Relevant data, such as smoking habits (35.7%) and immunomodulatory treatments, are included in Supplementary Table S1: methotrexate (71.3%), conventional synthetic disease-modifying antirheumatic drugs (64.3%), TNF antagonists (21.4%), and other biologics (35.7%). Further analysis of these variables was beyond our study’s scope.

It is crucial to recognize that LUS findings—such as B-lines and pleural irregularities—are not specific to ILD. Therefore, each patient underwent a comprehensive clinical evaluation, including appropriate diagnostic tests, to exclude other respiratory conditions like heart failure, infection, pleural effusion, atelectasis, asthma, or chronic obstructive pulmonary disease. However, some LUS features can help distinguish ILD from other lung disturbances such as the non-predominance of B-lines in the gravity-dependent regions of the lungs or their persistence despite diuretic therapy, the coexistence of B-lines and pleural line fragmentation, and the absence of pleural effusions. Furthermore, the same need for thorough clinical evaluation focused on differential diagnosis can also apply for golden standard HRCT.

Contrary to the concern raised, we believe that the extensive experience of our sonographers enhances our study’s reliability. Our findings show that LUS reliability is high when performed by adequately trained experts, underscoring the need for proper operator training—a standard not yet universally defined—as strongly recommended by recent consensus [4].

We also highlight our robust reliability results in assessing pleural line irregularities. This area has become a focus of recent LUS research, though consensus is lacking. Unlike B-lines, pleural line changes are not artifacts and can be evaluated more easily due to their superficial location and minimal interference from patient characteristics. Linear probes, commonly available in rheumatology clinics, are suitable for this purpose. Prospective research may clarify whether artificial intelligence will further enhance assessment, but our current semiquantitative method appears feasible for bedside use, consistent with our experience using OMERACT synovitis scores [6]. Further research areas include the usefulness of LUS as a screening test for detecting ILD abnormalities in asymptomatic RA patients, guiding the indication of HRCT, and as a safe monitoring tool of already diagnosed RA-ILD patients between their serial chest HRCTs.

In conclusion, our pilot study demonstrates substantial to excellent intra- and inter-explorer reliability of LUS when conducted by expert sonographers, highlighting the necessity of comprehensive training before clinical implementation. Despite certain limitations, our findings warrant further investigation in larger cohorts with disease severity stratification.

Evaluating pleural line abnormalities holds significant promise, yet consensus on imaging protocols and validation is essential before LUS can be widely adopted in clinical settings. We agree that LUS should not be seen as a standalone diagnostic tool for RA-ILD, but rather as a support method to guide the selection of patients for HRCT, underscoring that its findings must always be interpreted within the patient’s clinical context. We humbly consider that the reproducibility of LUS findings within the same or across different operators is an initial step in paving the way for its further validation as a screening, diagnosis, or monitoring tool.

We thank the authors again for their constructive comments, which have enriched the ongoing discussion and added clarity to our original manuscript.

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
