# Peer review of "Reply to Gomez-Rojas, M.A.; Tung-Chen, Y. Comment on “Vicente-Rabaneda et al. A Proposal for a New Lung Ultrasound Score in Rheumatoid Arthritis: The Reliability of Lung Ultrasound for Rheumatoid Arthritis-Associated Interstitial Lung Disease Diagnosis. J. Clin. Med. 2025, 14, 3701”"

_jcm, 2025, doi:10.3390/jcm14228119_

Round 1
Reviewer 1 Report (Previous Reviewer 1)
Comments and Suggestions for Authors
In their reply to the comment by Tung-Chen et al., the authors address the several considerations raised by the commenters, clarifying the limitations of their research methodology but also the importance of their results regarding the intra- and inter-observer reproducibility of the lung ultrasound (LUS) findings in the investigation of rheumatoid arthritis (RA)-associated interstitial lung disease (ILD). The key issue that has to be clearly delineated and prioritised for future research is the distinction of LUS as a screening vs. diagnostic test for RA-ILD. The authors are thus advised to emphasise further on:
1) The differential role of LUS as a screening, diagnostic, or monitoring imaging modality in probable RA-ILD. The gold standard test for the diagnosis of RA-ILD is the chest high-resolution computed tomography (HRCT). However, LUS could be useful as a screening test for detecting ILD abnormalities in asymptomatic RA patients and for guiding their further investigation by means of chest HRCT. Also, LUS could serve as an optimal tool for regular monitoring of already diagnosed RA-ILD patients between their serial chest HRCTs, as it does not expose them to ionising radiation.
2) The absence of validation of the LUS findings by comparing them with chest HRCT findings in all patients. As they correctly mention, they focused on the precision, i.e., reliability, of LUS rather than its accuracy, i.e., validity, to reveal RA-ILD changes. It is important to clarify that they eventually did not confirm or exclude the presence of RA-ILD in the enrolled patients.
3) The potential of LUS-depicted interstitial syndrome, even as a non-specific finding, to highly suggest the presence of ILD in the presence of particular sonomorphological features and in the right clinical context, e.g., the non-predominance of B-lines in the gravity-dependent regions of the lungs, their persistence despite diuretic therapy, the coexistence of pleural line fragmentation, and the absence of pleural effusions.
The work by Di Matteo et al. is an important first step in determining the value of LUS in the investigation of RA-ILD, as they showed the reproducibility of LUS findings within the same or across different operators, thus paving the way for its further validation as a screening, diagnostic, or monitoring tool. This would be useful to be clearly stated.
Author Response
COMMENTS BY REVIEWER 1
In their reply to the comment by Tung-Chen et al., the authors address the several considerations raised by the commenters, clarifying the limitations of their research methodology but also the importance of their results regarding the intra- and inter-observer reproducibility of the lung ultrasound (LUS) findings in the investigation of rheumatoid arthritis (RA)-associated interstitial lung disease (ILD). The key issue that has to be clearly delineated and prioritised for future research is the distinction of LUS as a screening vs. diagnostic test for RA-ILD. The authors are thus advised to emphasise further on:
1) The differential role of LUS as a screening, diagnostic, or monitoring imaging modality in probable RA-ILD. The gold standard test for the diagnosis of RA-ILD is the chest high-resolution computed tomography (HRCT). However, LUS could be useful as a screening test for detecting ILD abnormalities in asymptomatic RA patients and for guiding their further investigation by means of chest HRCT. Also, LUS could serve as an optimal tool for regular monitoring of already diagnosed RA-ILD patients between their serial chest HRCTs, as it does not expose them to ionising radiation.
2) The absence of validation of the LUS findings by comparing them with chest HRCT findings in all patients. As they correctly mention, they focused on the precision, i.e., reliability, of LUS rather than its accuracy, i.e., validity, to reveal RA-ILD changes. It is important to clarify that they eventually did not confirm or exclude the presence of RA-ILD in the enrolled patients.
3) The potential of LUS-depicted interstitial syndrome, even as a non-specific finding, to highly suggest the presence of ILD in the presence of particular sonomorphological features and in the right clinical context, e.g., the non-predominance of B-lines in the gravity-dependent regions of the lungs, their persistence despite diuretic therapy, the coexistence of pleural line fragmentation, and the absence of pleural effusions.
The work by Di Matteo et al. is an important first step in determining the value of LUS in the investigation of RA-ILD, as they showed the reproducibility of LUS findings within the same or across different operators, thus paving the way for its further validation as a screening, diagnostic, or monitoring tool. This would be useful to be clearly stated.
RESPONSE TO REVIEWER 1
We thank the reviewer for the favorable comments about our manuscript by Vicente-Rabaneda et al. as an important first step in determining the value of LUS in the investigation of RA-ILD based on showed reproducibility of LUS findings within the same or across different operators, thus paving the way for its further validation as a screening, diagnosis, or monitoring tool. We also appreciate the respectful comments about our Reply to the Letter sent by Di Matteo et al. about our manuscript.
Regarding the points we have been advised to emphasize further, we have made the following changes:
1) We thank the reviewer for pointing out that we should make a special comment about the usefulness of LUS as a screening test for detecting ILD abnormalities in asymptomatic RA patients, guiding the indication of HRCT, and as an optimal tool for regular monitoring of already diagnosed RA-ILD patients between their serial chest HRCTs, due to its safety.
We have included the following comments in page number 2, paragraph 6 and lines 93 – 96:
Further research areas include the usefulness of LUS as a screening test for detecting ILD abnormalities in asymptomatic RA patients, guiding the indication of HRCT, and as a safe monitoring tool of already diagnosed RA-ILD patients between their serial chest HRCTs.
2) We thank the reviewer for having identified such an important aspect to be clarified. We regret not having made it clear that we did confirm or exclude the presence of RA-ILD in all patients using HRCT.
We have, accordingly, clarified this key point in page number 2, paragraph 2 and lines 60 – 63:
Nevertheless, the presence of RA-ILD was confirmed or excluded in all patients by HRCT. Additionally, the diagnostic performance of the proposed LUS scores was evaluated, showing sensitivity, specificity, cut-offs and area under the receiver operating characteristic curve (AUROC) in Table 5.
3) We agree with this comment and, therefore, we have specified further the sonomorphological features that help distinguish ILD from other lung disturbances in page number 2, paragraph 4 and lines 75 – 78:
However, some LUS features can help distinguish ILD from other lung disturbances such as the non-predominance of B-lines in the gravity-dependent regions of the lungs or their persistence despite diuretic therapy, the coexistence of B-lines and pleural line fragmentation, and the absence of pleural effusions.
4) As we have previously mentioned, we thank the reviewer for the favorable comments about our manuscript by Vicente-Rabaneda et al. as an important first step in determining the value of LUS in the investigation of RA-ILD based on showed reproducibility of LUS findings within the same or across different operators, thus paving the way for its further validation as a screening, diagnosis, or monitoring tool.
We have, accordingly, included a comment in page number 3, paragraph 2 and lines 107 – 109:
We humbly consider that the reproducibility of LUS findings within the same or across different operators is an initial step in paving the way for its further validation as a screening, diagnosis, or monitoring tool.
Reviewer 2 Report (Previous Reviewer 2)
Comments and Suggestions for Authors
Authors provide a clear response with adressing all points raised in the previous comments. It is important that they clarify that asymptomatic individuals were also included.
I particularly appreciate the emphasis on pleural line abnormalities as true structural findings rather than artifaxt. LA in RA-ILD should not be viewed solely as an artifact based tehnique as the plaura line should be evolved further in research. Authors further add that B-lines and pleural changes are not specific for RA-ILD and that clinical context needs to be known. However that should not be limitation of LUS as the same can apply for golden standard HRCT.
Overall, the reply is clear, scientifically sound, and balanced.
Author Response
COMMENTS BY REVIEWER 2
Comments and Suggestions for Authors
Authors provide a clear response with adressing all points raised in the previous comments. It is important that they clarify that asymptomatic individuals were also included.
I particularly appreciate the emphasis on pleural line abnormalities as true structural findings rather than artifaxt. LA in RA-ILD should not be viewed solely as an artifact based tehnique as the plaura line should be evolved further in research. Authors further add that B-lines and pleural changes are not specific for RA-ILD and that clinical context needs to be known. However that should not be limitation of LUS as the same can apply for golden standard HRCT.
Overall, the reply is clear, scientifically sound, and balanced.
RESPONSE TO REVIEWER 2
We thank the reviewer for the favorable comments about our research.
We agree with the suggestions made and, accordingly, we have included the following sentences to further clarify our reply:
1) Regarding “clarify that asymptomatic individuals were also included“, we have made changes in page number 2, paragraph 2 and lines 54 – 55:
Importantly, as reflected in Supplementary Table 1, some of the enrolled patients were asymptomatic, with suspicion arising from abnormal LFTs or imaging, that were performed due to the presence of risk factors of ILD.
2) With respect to “However that should not be limitation of LUS as the same can apply for golden standard HRCT.”, we have added a comment in page number 2, paragraph 4 and lines 78 – 80:
Furthermore, the same need of thorough clinical evaluation focused on differential diagnosis can also apply for golden standard HRCT.
This manuscript is a resubmission of an earlier submission. The following is a list of the peer review reports and author responses from that submission.
Round 1
Reviewer 1 Report
Comments and Suggestions for Authors
In their commentary, the authors address several aspects of the original article by Di Matteo et al, which are important to consider regarding the value of lung ultrasound (LUS) in the diagnosis of rheumatoid arthritis (RA)-associated interstitial lung disease (ILD). Their arguments and considerations are clearly reported and well justified. The authors are advised to consider the further inclusion and/or elaboration of the following two points, which could provide additional value in this commentary:
1) The differential role of LUS in the screening for RA-ILD in RA patients without respiratory manifestations versus in the diagnosis of RA-ILD in RA patients with respiratory manifestations. While chest high resolution computed tomography (HRCT) is an irreplaceable imaging modality for the diagnosis of RA-ILD in suspect RA cases, LUS could have a more important role in the screening of asymptomatic RA patients for RA-ILD and their further referral for chest HRCT if relevant abnormal findings are depicted by LUS.
2) While the presence of B-lines per se is indeed a highly non-specific LUS finding for ILD, as correctly commented in lines 76-80, the particular characteristics of the ultrasonographic interstitial syndrome could improve the specificity of B lines for ILD, especially in the right clinical context, e.g., their non-predominance in the gravity-dependent regions of the lungs, their persistence despite diuretic therapy, the coexistence of pleural line fragmentation, the absence of pleural effusions.
Reviewer 2 Report
Comments and Suggestions for Authors
Dear authors,
The manuscript is written well with clear and balanced comments on the article "A proposal for new Lung....". You have identified relevant methodological limitations of the study (small sample size, combination of monocentric case series with systematic review, operator dependency of LUS...). Especially the exclusion of asymptomatic patients which would be the most important group to study considering the nature of RA-ILD.